# Research Trends in the Use of Remote Sensing for Inland Water Quality Science: Moving Towards Multidisciplinary Applications

**Simon N. Topp** [1,*] ![], **Tamlin M. Pavelsky** [1], **Daniel Jensen** [2,3] ![], **Marc Simard** [2] and **Matthew R. V. Ross** [4] ![]

1. Department of Geological Sciences, University of North Carolina at Chapel Hill, 104 South Rd, Mitchell Hall, Chapel Hill, NC 27599, USA; pavelsky@email.unc.edu
2. NASA Jet Propulsion Laboratory, 4800 Oak Grove Dr., Pasadena, CA 91109, USA; daniel.j.jensen@jpl.nasa.gov (D.J.); marc.simard@jpl.nasa.gov (M.S.)
3. Department of Geography, University of California at Los Angeles, Los Angeles, CA 90095, USA
4. Department of Ecosystem Science and Sustainability, Colorado State University, Fort Collins, CO 80523, USA; mrvr@rams.colostate.edu
* Correspondence: sntopp@live.unc.edu; Tel.: +1-303-917-2694

**Abstract:** Remote sensing approaches to measuring inland water quality date back nearly 50 years to the beginning of the satellite era. Over this time span, hundreds of peer-reviewed publications have demonstrated promising remote sensing models to estimate biological, chemical, and physical properties of inland waterbodies. Until recently, most of these publications focused largely on algorithm development as opposed to implementation of those algorithms to address specific science questions. This slow evolution contrasts with terrestrial and oceanic remote sensing, where methods development in the 1970s led to publications focused on understanding spatially expansive, complex processes as early as the mid-1980s. This review explores the progression of inland water quality remote sensing from methodological development to scientific applications. We use bibliometric analysis to assess overall patterns in the field and subsequently examine 236 key papers to identify trends in research focus and scale. The results highlight an initial 30 year period where the majority of publications focused on model development and validation followed by a spike in publications, beginning in the early-2000s, applying remote sensing models to analyze spatiotemporal trends, drivers, and impacts of changing water quality on ecosystems and human populations. Recent and emerging resources, including improved data availability and enhanced processing platforms, are enabling researchers to address challenging science questions and model spatiotemporally explicit patterns in water quality. Examination of the literature shows that the past 10–15 years has brought about a focal shift within the field, where researchers are using improved computing resources, datasets, and operational remote sensing algorithms to better understand complex inland water systems. Future satellite missions promise to continue these improvements by providing observational continuity with spatial/spectral resolutions ideal for inland waters.

**Keywords:** remote sensing; water quality; lakes; rivers; inland waters; scientific advancement

## 1. Introduction

Remote sensing has long been promised as a tool for large-scale monitoring of inland water quality. Dating back to the early 1970s, airborne and satellite sensors have been used to examine a wide range of water quality constituents [1,2]. In the 50 years since, scientists have produced hundreds of peer-reviewed publications presenting models estimating biological, chemical, and physical properties

of complex waterbodies (see reviews [3–6]). Despite this proliferation of publications, existing reviews focus almost exclusively on methodological approaches rather than on the scientific contributions of remote sensing to our understanding of water quality, so characterization of the extent to which remote sensing has improved our knowledge of inland water dynamics remains limited.

The historical tendency of inland water remote sensing to focus largely on methods development (here defined as data collection and processing and/or algorithm calibration and validation), contrasts starkly with that of related fields in terms of both the scope of research questions and the scale of studies. For terrestrial remote sensing, algorithm development throughout the 1970s (e.g., Normalized Difference Vegetation Index (NDVI) [7]) led to publications focused on spatially expansive, complex processes as early as the mid-1980s. These papers include studies on global land use [8], global vegetation analysis [9], and connections between primary productivity and carbon cycling [10,11]. For ocean color remote sensing, early methods development led to global datasets and estimations of oceanic primary productivity by the late 1980s [12,13]. Comparatively, global data products for inland water quality are limited, with a few key exceptions (e.g., [14]) despite widespread acknowledgement of their importance from the inland water scientific community [15,16]. This slow evolution can be partially explained by well-known challenges related to remote sensing of complex waterbodies, as well as the limited availability of sensors appropriate for inland water quality remote sensing [17], discussed in detail with other challenges in Section 7.

Previous reviews have provided excellent summaries of the technical approaches available to retrieve inland water quality parameters through remote sensing, as well as the current limitations of the field [3–6,18–20]. Instead, we focus not on methodological details, but on the overall purpose and impact of past publications, how those impacts have changed over time, and how the field may evolve in the future. We quantify broad-scale trends through bibliometric analysis of search engine results. A subset of the most relevant published papers (n = 236) was identified using existing reviews, citation counts, database queries, and journal-specific searches. The identified papers were subsequently read, with key attributes documented in order to analyze trends and patterns in methodological approaches, model application, research focus, and study scale over time. Here, trend refers to a pattern with directionality over time or space. We limit our analysis to airborne and satellite remote sensing publications focusing on lakes, rivers, deltas, and estuaries, although we fully acknowledge that these publications were preceded by years of vital methods development using handheld and shipborne sensors (e.g., [21–26]). Similarly, given the focus of this paper on the remote sensing of lake, river, delta, and estuarine systems, which present their own unique challenges [17], we excluded studies on near shore ocean waters and the Laurentian Great Lakes due to their similarity to Ocean Color Remote Sensing where well established methods already exist.

Our results highlight a nearly 30 year period focusing predominantly on methods development prior to a spike in publications, beginning in the early 2000s, applying well validated algorithms to identify spatiotemporal trends, drivers, and impacts of changing inland water quality on ecosystem functions and human populations. Study scale exhibits a similar trend towards increasingly large areas with more waterbodies studied over longer periods of time, slowly moving closer to regional and global-scale data products. Through both broad and detailed inspection of the field, our results suggest that the past decade of inland water remote sensing has led to a fuller understanding of inland water processes by focusing on challenging science questions and increased study scales. This contribution continues today with an ever-expanding body of available data, processing platforms, and methodologies.

We contextualize our analysis of the literature by: (1) summarizing the primary water constituents measured with earth observation instruments, (2) providing a brief overview of common modelling approaches to measure those constituents, and (3) discussing the limitations that have hampered past research. This contextual information is followed by the bibliometric and index analysis described above. We conclude with a discussion of potential future directions for the field.

## 2. Earth Observation Sensors and Optically Active Waterbody Constituents

The work reviewed here focuses primarily on passive optical satellite sensors capable of large-scale remote sensing research. In general, these are either ocean color sensors such as the Moderate Resolution Imaging Spectroradiometer (MODIS) [27–30], the Medium Resolution Imaging Spectrometer (MERIS) [31–34], and the Sentinel-3 Ocean and Land Cover Instrument (OLCI) [35], or land surface optical sensors including the Landsat series (Multispectral Scanner (MSS): [36–38]; Thematic Mapper (TM): [39–41]; Enhanced Thematic Mapper (ETM+): [42–44]; and Operational Land Imager (OLI): [45, 46]), Sentinel 2 A/B MultiSpectral Instrument (MSI) [47,48], and SPOT High-Resolution Geometric Sensor (HRG) [49]. A subset of researchers have used high-resolution commercial sensors including WorldView 2 [50] and IKONOS [51,52]. The above sensors vary significantly in their applicability, based largely on their spatial, temporal, spectral, and radiometric resolutions. Temporal and spatial resolutions determine the scale of processes that can be captured by a given sensor. In general, land surface sensors have a finer spatial resolution (~10–30m) but coarser temporal resolution (~1–2 weeks), allowing them to detect spatial patterns in water quality in smaller waterbodies (e.g., small lakes and rivers) but with only 1–2 observations per month depending on the sensor and cloud cover conditions. Comparatively, ocean color sensors are characterized by coarse spatial resolutions (~300–1000 m) but finer temporal resolutions (~daily), limiting observations to large waterbodies but facilitating examination of processes that occur at short timespans. A more in-depth discussion on the effects of varying resolutions across ocean and terrestrial sensors can be found in Olmanson, Brezonik, and Bauer (2011) [53] and the Committee on Earth Observation Satellites (CEOS) (2018) [54]. Additionally, technical discussions and summaries of the spatial, temporal, spectral, and radiometric resolutions of the above sensors are provided by Gholizadeh et al. (2016) [3] and Matthews (2011) [4].

Since water is highly absorptive within the near and shortwave infrared spectrum, the majority of water-leaving radiance occurs within the visible spectrum with slight variations dependent on temperature and salinity [55,56]. The primary exception is in optically complex waters (due to high turbidity and/or bottom reflectance), where sediment reflectance exceeds the absorptive properties of water in the near and shortwave infrared wavelengths [49,57]. Relatively high absorption within the visible spectrum leads to a low range of reflectance values when compared to land surface remote sensing. This low range requires high sensitivity (i.e., high radiometric resolution) to detect small changes in reflectance [4]. Different concentrations of varying water quality parameters lead to various absorption features and backscatter peaks within the water leaving radiance. The spectral resolution, measured by the range of wavelengths captured by individual sensor bands, needs to be sufficiently fine to capture spectral peaks and accurately estimate the contribution of a given water quality parameter to the overall spectral signature [17]. The sensors mentioned above are all multispectral sensors, meaning that they have a small number of relatively wide bands (~10 nm to ~80 nm) placed within the visible to mid-infrared spectrum. These coarse bandwidths can complicate retrieval of water quality parameters [17]. In order to better capture the specific absorption features and backscatter peaks within a waterbody's spectral signature, a subset of publications have utilized hyperspectral sensors that provide hundreds of narrow (1–10 nm), contiguous bands spanning the visible to shortwave infrared spectrum (see [58]). Currently, the majority of hyperspectral sensors are airborne or in planning stages for future satellite missions [59]. Within inland water remote sensing, applications of hyperspectral remote sensors include the use of Hyperion [60,61], the Compact Airborne Spectrographic Imager (CASI) [62,63], the Airborne Prism Experiment (APEX) [64], and NASA's HyMAP scanner [65], Airborne Visible/Infrared Spectrometer (AVIRIS) [66], Airborne Visible/Infrared Spectrometer-Next Generation (AVIRIS-NG) [67], and Portable Remote Imaging Spectrometer (PRISM) [68].

Regardless of sensor, the optically active water parameters that contribute to the total water-leaving signal are phytoplankton, organic and inorganic suspended solids, and colored dissolved organic matter (CDOM) [57,69,70] (Table S1). The sum of these three individual constituents, in combination, attribute to differences in overall water clarity, which is frequently used as a proxy for water quality [15,71]. Publications leveraging relationships between optically inactive constituents, which have no detectable

spectral signal, and the optically active constituents listed above have provided remote sensing models for nitrogen and phosphorous [72–74], dissolved oxygen [39,47], and heavy metals [68,75]. However, compared to optically active parameters, these optically inactive constituents require site specific algorithms due to varying regional correlations with optically active water quality constituents. Publications examining the remote sensing of inactive constituents date back to the early 90s [40,76]; however, they appear relatively infrequently within the literature and are not discussed in detail here. Below, we describe the optically active constituents with their distinct spectral signatures.

*2.1. Chlorophyll-A*

Chlorophylls are the photosynthetically active compounds that convert light into energy for photosynthesis. Remote sensing studies primarily focus on chlorophyll-a (chl-a), which is the most abundant chlorophyll and is present within all plants, algae, and cyanobacteria that photosynthesize. In aquatic systems, it is used as a proxy measure of total algal biomass [77]. The algal biomass of a waterbody controls its overall biological productivity, also known as trophic state, making it an ideal indicator of ecosystem integrity [78,79]. While not all algal blooms are inherently harmful, blooms containing certain species, most commonly phycocyanin-producing cyanobacteria, are toxic to humans, livestock, and wildlife [80]. Anthropogenically driven nutrient loading and climate change in recent decades have increased the size and frequency of these harmful algal blooms worldwide [81].

Optically, the spectral signature of chl-a varies depending on its concentration in relation to other water quality parameters and the composition of phytoplankton phenotypes producing the signal [82,83]. For low biomass, oligotrophic to mesotrophic waterbodies, the chl-a spectrum is characterized by a sun-induced fluorescence peak around 680 nm [84–86]. For high biomass, eutrophic waterbodies, the florescence signal is masked by absorption features and backscatter peaks centered at 665 nm and 710 nm respectively [87]. The ratio between these two wavelengths has been used to accurately estimate chl-a concentrations in numerous studies [88–90]. Beyond basic constituent retrieval, research focusing on chlorophyll includes the detection of harmful cyanobacteria [91–93] and phycocyanin [62,94], assessment of trophic state [44,45,95], and algal bloom development and dispersion modelling [96–99].

*2.2. Total Suspended Solids*

Total suspended solids (TSS) refers to both inorganic and organic particles held in suspension throughout a water column. Controls on the composition of organic and inorganic particles vary geographically, with some areas driven primarily by inorganic sediments and others by phytoplankton. In the literature it is referred to variously as total suspended matter, suspended sediment concentration, and particulate matter, though the precise definitions of these terms sometimes vary. Monitoring TSS fluxes has strong implications for biogeochemical cycling in terms of nutrient transport [100], heavy metal loading [101], light conditions [102], and global carbon budgets [103]. Terrestrial carbon deposition into lakes and reservoirs, largely in the form of TSS, is double that of deposition into the ocean [104,105], despite lakes comprising only 3%–3.7% of the total land area [106,107]. Simultaneously, the settling out of TSS into lake bottom sediments provides a carbon sink, with current global carbon sequestration estimates ranging from 0.06–0.27 Pg year$^{-1}$ [103,105]. On a local scale, high TSS reduces light penetration through increasing turbidity and leads to benthic smothering, impacting species composition and primary productivity from macrophytes [108,109]. Finally, TSS concentrations and flux in rivers capture the landscape processes controlling delivery of erosional products from land to ocean [110,111].

Spectral signatures of TSS concentrations can vary significantly based on the particle size and composition of organic to inorganic materials [112,113]. Organic-dominated systems derive their spectral signatures from algae concentrations and can share the pronounced absorption features and backscatter peaks described above for chlorophyll [114]. As inorganic TSS concentrations increase within a waterbody, the location of the spectral maximum moves from around 550 nm into the red or

near-infrared wavelengths [49] with waterbody specific variation dependent on chlorophyll and CDOM concentrations. Remote sensing studies examining TSS focus largely on riverine and coastal systems, with notable studies including estimates of TSS delivery to the ocean [110], variability in sediment plume size [28,115,116], impacts of reservoirs on sediment concentration [117], impacts of land use change on sediment delivery [118], and variability of sediment in lagoons [119]. TSS concentrations can be correlated with various optically inactive water quality parameters and have subsequently been used to infer the concentration of phosphorous [3], mercury [118], and other metals [65] at local scales.

## 2.3. Colored Dissolved Organic Matter

Colored (or 'chromophoric') dissolved organic matter is the colored portion of total dissolved organic carbon. Sources of CDOM can be either autochthonous (i.e., phytoplankton) or allochthonous (i.e., terrestrial carbon). Of the two sources, allochthonous carbon leached out of surrounding soils is generally the dominant control of total lake and river dissolved organic carbon [120]. Photo and biodegradation of CDOM can contribute to elevated levels of $CO_2$ within lacustrine systems [121]. Recent studies of $CO_2$ concentrations in Chinese [122] and US [123] lakes found that ~60%–70% were supersaturated with $CO_2$. Globally, this oversaturation leads to 0.35–0.43 Pg year$^{-1}$ of carbon off-gassed into the atmosphere, in addition to an estimated 1.8 Pg year$^{-1}$ emitted from streams and rivers [124]. At low levels, CDOM absorbs harmful ultraviolet radiation with minimal impact on light penetration within the visible spectrum [125]. As concentrations increase, absorption of low-wavelength light by CDOM regulates the light availability of primary producers, controlling net productivity and trophic structure [125,126]. Continued monitoring of CDOM directly, and as a proxy for total dissolved organic carbon, provides a better understanding of carbon inputs and processing in freshwater systems.

Highly absorptive in the visible spectrum, elevated levels of CDOM lead to stratified, dark waterbodies with limited light penetration [127]. Similar to TSS, the reflectance spectra of waterbodies with varying concentrations of CDOM are highly dependent on the composition of other optically active constituents, and in certain areas can be complicated by the presence of colloidal iron, which shares similar optical properties [128]. CDOM's contribution to water-leaving radiance is characterized by an exponential increase in absorption as wavelength decreases [129]. Intuitively, this would suggest that CDOM models should incorporate wavelengths in the blue spectrum; however, excessive absorption by CDOM and low natural water-leaving radiance at low wavelengths reduces the usable signal [24,130]. As a result, algorithms commonly incorporate a green/red ratio (e.g., [47,131–133]). Remote sensing studies focusing on CDOM range in application from identifying trends in inland water carbon content [134,135] to examining landscape-level drivers of CDOM distributions [50,136]. Work in rivers highlights controls of carbon export in arctic landscapes [137] and relationships governing CDOM variation in river estuaries along with the resulting impact on correlated concentrations of methylmercury [68]. An in depth review of CDOM and its optical properties was published by Coble (2007) [138].

## 2.4. Water Clarity

The combination of chlorophyll, suspended sediments, and CDOM collectively contributes to overall water clarity. Most commonly, Secchi Disk depth or turbidity are used as relative measures of clarity. The former metric, developed more than 150 years ago, quantifies the maximum visible depth of a white and black disk lowered into a waterbody [139,140]. In comparison, turbidity is an explicit measurement of light scattering within a water column caused by suspended and dissolved particles. Water clarity regulates freshwater ecosystems through light attenuation and control over epilimnion depth [141]. Numerous studies have examined the role of water clarity in thermal stratification [142,143], lake metabolism [144,145], and biodiversity [108]. Generally, a shallower thermocline and reduced light penetration associated with degraded water clarity reduces photosynthesis of submerged macrophytes and other primary producers [108,146].

Remote sensing retrievals of water clarity almost universally use wavelengths and band ratios that include the red spectrum in some way (e.g., [71,76,147–152]). Reflectance at these wavelengths accounts for total sediment and chlorophyll concentrations such that increasing brightness is associated with decreased water clarity [4]. Water clarity has long been acknowledged as a proxy for nutrient availability and chlorophyll concentrations within lakes [153–155]; as a result, remote sensing studies frequently use it as a proxy for overall lake trophic status (oligotrophic, mesotrophic, or eutrophic) [95,156,157].

## 3. Modelling Approaches

Models that leverage the relationship between a waterbody's optical qualities and its concentration of optically active water quality constituents are commonly referred to as bio-optical algorithms [153]. In inland waters, these models can be categorized as empirical, semi-analytical, or machine learning based (Table S2). While inherently empirical, we distinguish machine learning techniques separately due to their computational complexity and ability to handle non-linear relationships. As discussed below, all three of these modeling approaches have benefits and shortcomings in terms of applicable scale, model transparency, and model complexity.

### 3.1. Empirical Models

The most common approach to inland water remote sensing involves fitting a standard linear regression between spectral band/band ratio values and temporally coincident in situ water quality measurements. One inherent limitation of this approach is its non-generalizability across large spatial and temporal scales where variations in atmospheric and water composition create large variability in observed spectral signatures of water quality parameters. As such, empirical models are restricted to confident predictions only within the range and setting of the input data. This restriction limits their application across spatiotemporal domains. At a local scale, empirical modelling accounts for the site-specific optical qualities of the water, but with increasing spatial or temporal scales, optically non-homogenous waterbodies and changing atmospheric conditions complicate parameterization [158]. These shortcomings are often outweighed by the benefits of model transparency, simplicity, and minimal computational requirements.

The family of empirical models can be split into purely empirical and semi-empirical approaches. The purely empirical approach derives relationships using input band and band ratio values as coefficients, often generating multiple models and choosing the best fit through comparison of error metrics. Purely empirical approaches date back to the 1970s and 80s, with notable applications examining trophic state in Wisconsin [2] and Minnesota [73], and turbidity and chlorophyll in Australian lakes [159].

In contrast, semi-empirical models use multi-band index values with some basis in the physical properties of the constituent of interest. These models largely focus on the measurement of water clarity, chl-a, cyanobacteria, and TSS. Like terrestrial vegetation indices (e.g., NDVI), they are designed to enhance the spectral properties of the constituent of interest while reducing noise from extraneous optical parameters; however, unlike semi-analytical approaches (described below), semi-empirical models don't incorporate any inverse modelling of the inherent optical properties of a given waterbody. Notable semi-empirical indexes include the normalized difference chlorophyll index [160], the maximum chlorophyll index [161], the Floating Algal Index [162], and the normalized difference suspended sediment index [163]. Application of these semi-empirical indexes has contributed to robust algal bloom detection [164], determining the presence of harmful cyanobacteria concentrations associated with eutrophication [82], and modelling sediment concentrations in rivers and deltas [163]. Due to their basis in physical properties, semi-empirical models are more generalizable than purely empirical approaches. However, they necessitate measurements of specific wavelengths that capture absorption features and scattering peaks, restricting their applicability to sensors with suitably placed band centers and sufficient spectral resolution.

### 3.2. Semi-Analytical Models

Analytical and semi-analytical models are physics based and involve parameterization based on the inherent optical properties (IOPs) of water and the atmosphere, where IOPs refer to the optical properties of the medium of interest that are independent of the ambient light field [57,69]. The IOPs of a given waterbody are modelled in coordination with apparent optical properties (including illumination conditions, sensor orientation, and field of view) to construct theoretical absorption and backscattering values which can then be decomposed through an inverse equation to estimate optically active water quality constituents (described below) [18,57,70,165]. For purely analytical models, the inverse equation is parameterized based purely on light physics; however, these are rarely used for optically complex waters where the interactions of numerous water quality constituents become difficult to model. As a result, semi-analytical models, which incorporate in situ measurements to parameterize the inverse equation, are the primary form of physics based algorithms developed for inland water quality remote sensing retrievals [4]. This modelling approach evolved from the reflectance approximation developed by Morel and Prieur (1977) [166], who studied turbidity and chlorophyll in ocean waters. Compared to empirical and semi-empirical algorithms, semi-analytical models are mechanistic, make apriori assumptions regarding light physics, and are theoretically generalizable outside the range of a given study; however, the application of any single model to optically nonhomogeneous waterbodies requires large amounts of in situ validation data and remains challenging [16].

A prerequisite to this modelling approach is understanding the light physics that control reflectance as particle size, composition, and concentration vary. These properties are modelled through the absorption and backscattering coefficients of all the optically active constituents found within the study area (Equation (1)). While derivations of semi-analytical models come from numerous sources (e.g., [167–170]), the basic form of the preliminary equation follows Equation (1).

$$R(\lambda) = Y \times \frac{b(\lambda)}{a(\lambda) + b(\lambda)} \tag{1}$$

Here, the total reflectance just below the water's surface ($R$) at wavelength $\lambda$ is equal to the backscattering at the given wavelength over the absorption plus the backscattering at the given wavelength times an empirically or analytically derived constant $Y$. The absorption and backscattering coefficients can be further broken down into absorption and backscattering for each optically active constituent. (e.g., $b(\lambda) = b_{water}(\lambda) + b_{cdom}(\lambda) + (\lambda)b_{chl} + (\lambda)b_{tss}$). Values for $R$ are either generated through in situ measurements of reflectance and water quality or theoretically generated using physical modelling software such as HydroLight [171]. These generated spectral signatures are then used to parameterize an inverse model that decomposes $R$ into optically active constituent concentrations through their absorption and backscatter coefficients. One benefit of this inverse modelling procedure is the ability to estimate multiple water quality parameters simultaneously. However, model development is inherently complicated and, depending on if atmospheric corrections have been applied, requires information about atmospheric composition, bottom reflectance, and extensive in situ sampling. Even so, the literature contains numerous examples of successful applications of semi-analytical models across large spatiotemporal scales. Early development of semi-analytical modelling for inland waters was led by researchers such as Dekker [26,172] and Kutser [173] examining chl-a, TSS, and CDOM. More recently, Heege et al. (2014) [174] developed a semi-analytical algorithm for turbidity across the Mekong Delta with strong validation results using MODIS, Landsat, and RapidEye, Lymburner et al. (2016) [175] applied a semi-analytical algorithm to a multi-decadal study of TSS in Australian lakes, and both Volpe et al. (2011) [119] and Zhou et al. (2017) [176] applied semi-analytical algorithms across multi- and hyper-spectral data to detect TSS in shallow lagoons. For a more detailed description of semi-analytical modelling, see Dekker, Vos, and Peters (2001, 2002) [177,178], Giardino et al. (2019) [18], Morel (2001) [165], and IOCCG (2000) [57].

*3.3. Machine Learning Models*

In recent years, increases in computational capacity and available data have created opportunities for novel approaches to data analysis. While inherently empirical, machine learning approaches are differentiated by their ability to operate in multidimensional space with complex non-linear relationships [179]. The spectrum of machine learning methods for remote sensing applications is broad [180,181]; here, we focus on the benefits and limitations of machine learning methods generally, along with some notable examples in the field of inland water remote sensing. A more detailed review of machine learning methodology for remote sensing was published by Lary et al. (2016) [181].

Within inland water remote sensing, machine learning algorithms including artificial neural networks [182–184], genetic algorithms/programming [185,186], support vector machines [187], random forest/boosted regression trees [188], and empirical orthogonal functions [189,190] have all shown promise in accurately estimating water quality parameters across a variety of spatiotemporal scales. As with traditional empirical models, machine learning approaches are only applicable within the range and setting of data used to train a given model. However, unlike traditional empirical models, most machine learning models use iterative learning to reduce overall error and maximize model fit [191]. Depending on the parameterization of the model and the amount of training data available, this approach may lead to over-fitting of the data, especially in models with numerous input variables subject to collinearity such as adjacent hyperspectral bands [192]. To avoid overfitting, machine learning methods require the provision of separate training and testing datasets that contain representative samples of the parameters of interest. The power and scalability of most machine learning algorithms is dependent on the quality and range of the training and testing data. Given the proper inputs, these algorithms can produce generalizable models that capture complex, non-linear relationships between remotely sensed reflectance and biogeophysical parameters. While modelling chl-a and turbidity in Lake Chagan, China, Song (2011) [183] found reductions in root mean square error of 76% and 65%, respectively, when comparing traditional regression techniques to artificial neural networks. Similarly, Xiang et al. (2015) [193] found a 20% increase in trophic state classification accuracy when using machine learning compared to multivariate regression.

## 4. Challenges and Limitations Within the Field

The literature reviewed here highlights that, despite the diverse modelling approaches discussed above, several barriers still exist that limit the progress of inland water remote sensing. Specifically, sensor design, atmospheric effects, dynamic waterbodies, and institutional barriers, all of which present legitimate challenges to increasing the scale and robustness of remote sensing algorithms. Here, we discuss these issues in detail to provide context on the limitations of the reviewed literature.

At the most basic level, many sensors are limited in the types of observations they can make. Multispectral, broad-band satellites like the Landsat TM/ETM+ series were engineered for terrestrial applications and lack the spectral resolution, band centers, and signal-to-noise ratios ideal for complex waters. Their relatively infrequent return periods make them more suited to detecting long-term changes as opposed to daily or weekly variation. Ocean color sensors including MODIS, SeaWiFS, and MERIS have higher spectral resolution and frequent return periods, but they lack the spatial resolution to capture narrower inland water bodies, particularly rivers (see [4,17] for detailed discussion). The newest generation of sensors has been designed to overcome some of these issues [18,19], though the limited precision of broad spectral bands remains a challenge. While they lack certain band centers useful for inland water remote sensing, new sensors such as the Landsat 8 Operational Land Imager (OLI) and the Sentinel 2 MultiSpectral Instrument (MSI) have increased signal to noise ratios, improved radiometric and temporal resolution, and aerosol-specific bands making them better equipped to handle the size and complexity of inland waters [194,195].

Regardless of sensor choice, among the largest barriers to remote sensing of inland waters is controlling for varying atmospheric effects. The signal to noise ratio of top-of-atmosphere radiance over waterbodies can vary substantially with different atmospheric water vapor and aerosol concentrations.

In order to accurately estimate water quality parameters, the atmospheric effects need to be controlled for through precise atmospheric corrections [20,196]. These corrections are particularly important over large spatiotemporal domains because atmospheric conditions can vary significantly. Historic correction procedures are largely based on open ocean remote sensing and assume zero water leaving radiance beyond the visible spectrum [197]. This assumption does not hold over optically complex waters where chlorophyll, suspended sediment, and bottom reflectance lead to true non-zero radiance in the near infrared. The result is an overestimation of aerosol thickness and an overcorrection of visible wavelengths in turbid waters [198]. Progress has been made improving atmospheric correction algorithms over complex waters through the use of radiative transfer functions [198], pseudo-invariant features [194], dark pixel extraction [199], and SWIR-based correction procedures [200,201]; however, many methods lack transferability between sensors making it difficult to compare surface reflectance products across platforms [202]. Atmospheric correction is further complicated by adjacency effects from surrounding land. Radiation reflected from relatively bright land is scattered by the atmosphere, increasing noise over adjacent, relatively dark waterbodies. Solving adjacency issues typically involves computationally expensive radiative transfer functions, though recent progress has been made using models that reduce computational requirements by approximating atmospheric scattering within the correction procedure [203].

Independent atmospheric challenges are exacerbated by the dynamic nature of waterbodies themselves. Changing water conditions and bio-fouling of in situ sensors can make it difficult to capture coincident field and satellite observations necessary for model development [204,205]. From a reflectance standpoint, algal mats, surface macrophytes, and sun glint (specular reflection of sunlight towards the sensor) all contribute extraneous signals to observed water-leaving radiance. The body of literature on these issues is significant, and processing schemes to isolate and/or remove these signals are continually improving. For sun glint, removal schemes can range from relatively simple empirical models such as those tested by Kutser, Vahtmäe, and Praks (2009) [206] to more complicated radiative transfer functions [207]. For algal mats and floating macrophytes, semi-empirical threshold-based algorithms including the Floating Algal Index [162], Maximum-Peak Height [87], and adaptations of classic NDWI indexes [92] have all provided robust delineation of water from adjacent algal and macrophyte signals. Additionally, varying sediment types between regions can affect the relationship between reflectance and measurements of TSS [113]. These variations can be partially accounted for using band ratio algorithms that are generalizable across sediment types [208].

The above technical barriers represent legitimate challenges to extracting water quality constituents from dynamic inland systems. However, existing retrospectives on the past 50 years in the field indicate that technical barriers alone are not responsible for the slow progress towards applying remote sensing as a tool to better understand inland water systems. Bukata (2013) [209] insightfully proposes that one explanation may be the relatively isolated nature of the field and the historic lack of collaboration with related ocean color remote sensing. This observation is supported by Downing (2014) [210], who describes the fields of oceanography and limnology as "twins, mostly separated since birth". This lack of institutional communication has had ripple effects, reducing collaborative projects and limiting funding sources. Collaboration is further reduced through the inherent scale of the research. Technical challenges with spatiotemporally expansive studies generally constrain inland water research efforts to localized scales. This pattern contrasts with ocean remote sensing, where international study areas have led to numerous, well-funded, multinational research efforts [211]. Communication and collaboration leading to these large research efforts is facilitated by international organizations like the International Ocean-Colour Coordinating Group (IOCCG). Recent work done by the IOCCG [20], as well as emerging groups like AquaWatch (https://www.geoaquawatch.org/), are working towards similar goals for inland water remote sensing researchers, but these efforts are still in their infancy compared to their ocean color counterparts. The nature of the field and its apparent lack of cohesion may, in part, stem from the fact that it is spread across many different disciplines. A Scopus search query of inland water quality remote sensing returns publications from over 350 distinct journals

spread across hydrology, ecology, biogeochemistry, environmental management, and engineering, indicating that much of the research is spread across different niches and sub-disciplines. However, the following review of the literature indicates that efforts to overcome these technical and institutional barriers have made significant progress in improving inland water quality remote sensing efforts, particularly over the past decade. Below we highlight this progress through an analysis of how remote sensing of inland water quality has been used to answer challenging science questions. Additionally, we contextualize these recent advances within the literature and discuss how researchers are working towards further addressing them in the future (Section 7).

## 5. Evolution of Inland Water Remote Sensing Publications

In order to analyze the progression of publications on remote sensing of inland water quality, we carried out two analyses: the first identifies general trends in publication patterns, while the second analyzes trends in modelling approaches and research focus.

### 5.1. Overarching Trends in the Field of Inland Water Remote Sensing

General trends were identified through a bibliometric analysis of search results from the Elsevier Scopus database (conducted July 2018). Database titles, keywords, and abstracts were searched for the terms 'remote sensing', 'water quality', and either 'lake', 'reservoir', 'river', 'delta', 'estuary', or 'inland waters' (along with variants, i.e., 'lake' and 'lakes'). The search results returned 1,186 distinct articles published in peer-reviewed journals dating back to 1970 (Figure 1). Bibliometric data were extracted from the query using the Bibliometrix package in R [212].

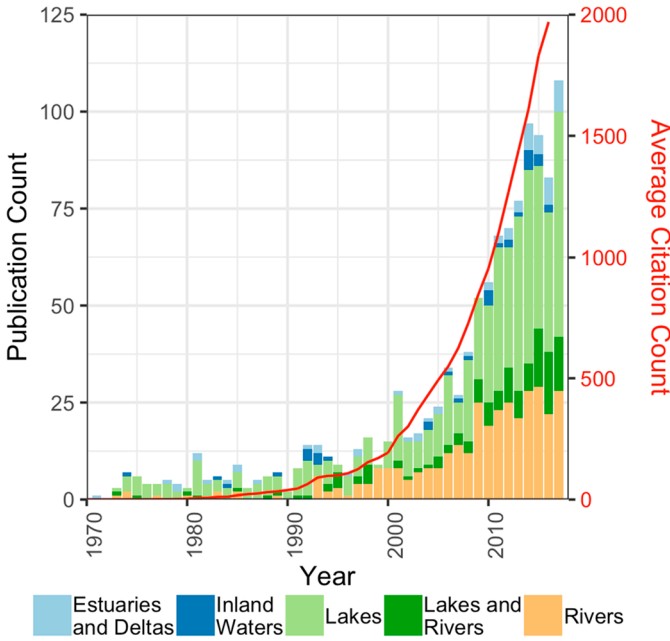

**Figure 1.** Published papers per year returned from Scopus search queries and grouped by search term. Average citation count is the sum of citations for all papers averaged over the number of years since their publication.

The results of the Scopus bibliometric analysis indicate that inland water quality remote sensing has been growing dramatically since its introduction in the 1970s. The annual average increase in publications over the study period is 8.9%, but examination of the trend indicates that it is best represented by a simple power law function ($R^2 = 0.848$), with a sharp increase in publications starting in the early 2000s. Power law functions allow for the calculation of a doubling time which represents the amount of time it takes a population to double in size starting from any given timepoint. Bornmann

and Mutz (2015) [213] calculated the doubling time and average annual growth rate for total academic publishing between 1980 and 2012 to be approximately 23.7 years and 2.96% respectively. For the same period, remote sensing of inland water quality grew at three times that rate, with a doubling time and average annual growth rate of 8.3 years and 10.01% respectively. The most pronounced year-on-year jump occurs right after 2008, which corresponds to the public release of freely available Landsat imagery by NASA and the US Geological Survey. After removing the overall trend of the power law function, a t-test on the residuals for the 5 years before and after 2008 indicates a significant increase in publications for the period after Landsat was made public (95% CI = 0.3–0.7, $p$ = 0.0016). This result is consistent with previous research showing that for multiple earth observation fields, the release of the Landsat archive resulted in more frequent and larger-scale studies [214].

Further analysis of the bibliometric data shows that while contributions to the literature come from a diverse set of sources, there are a few distinct countries, journals, and authors that are disproportionately active within the field. Publications from the United States and China are responsible for 26.1% and 21.4% of the total publications respectively (Figure 2). Similarly, while there are contributions to the literature from 3362 authors or co-authors, publications that include the top ten most productive authors comprise 17% of the total search results (Table 1). The cumulative contribution of publications from the top ten journals comprise nearly one third of the entire search. Of the 378 publications from these top ten journals, 60% are from strictly remote sensing journals. When expanded out to the entire query, 18% of the returned journals include a remote sensing term in their title and account for 33% of all publications. This pattern is worth noting for two reasons. First, remote sensing journals are more likely to contain methods development papers. Secondly, it suggests that many publications are focused primarily on communicating advances within the remote sensing community, with perhaps less outreach to hydrologists, ecologists, and other scientists not inherently focused on remote sensing.

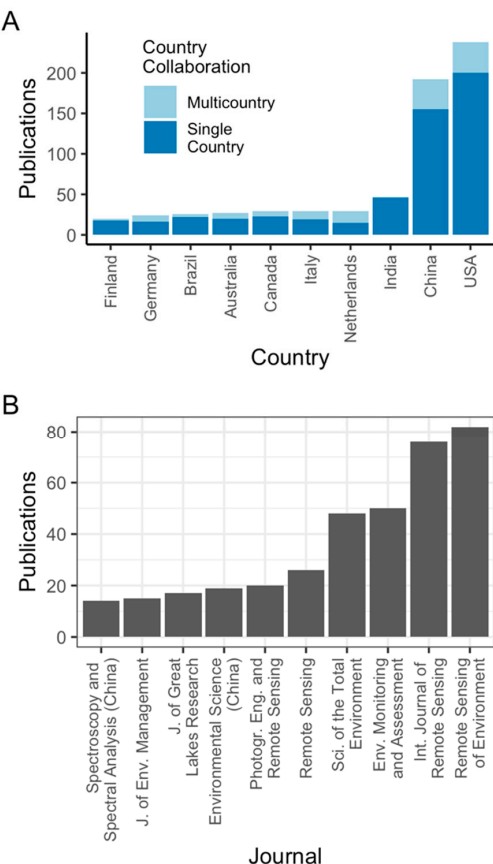

**Figure 2.** Distribution of publications returned from Scopus query for the top ten most productive countries (**A**) and top ten most published journals (**B**).

**Table 1.** Summary data from Scopus query for inland water quality remote sensing.

| Scopus Query Summary | |
|---|---|
| Total Publications | 1186 |
| Distinct Journals | 342 |
| Distinct Keywords (Scopus) | 7706 |
| Distinct Keywords (Authors) | 2447 |
| Average citations per publication | 16.6 |
| **Authorship Summary** | |
| Distinct Authors | 3,362 |
| Authors per Documents | 5.24 |
| **Contributions Summary** | |
| Contribution from top 10 Countries | 676 (54.8%) |
| Contribution from top 10 Authors | 209 (16.9%) |
| Contribution from top 10 Journals | 378 (30.6%) |

*5.2. Detailed Analysis of Literature Patterns and Scale*

In order to more deeply examine trends in remote sensing of water quality, we identified a subset of 236 papers within existing reviews [3–5] and from keyword searches containing common inland water remote sensing terms (e.g., combinations of 'remote sensing', 'lakes', 'rivers', 'chlorophyll', 'CDOM', 'TSS', and 'inland waters') in relevant databases (Article+, Google Scholar, Scopus, and Web of Science). Papers were chosen based on a combination of their search relevance, citation count, and subject focus. While we strived to be comprehensive in the inclusion of papers, some relevant studies were inevitably missed. We conducted more intensive journal-specific searches within high impact journals including Science, Nature, PNAS, WRR, Association for the Sciences of Limnology and Oceanography (ASLO) journals, and Ecological Society of America (ESA) journals to ensure the inclusion of studies that utilized remote sensing but focused more on scientific application of remote sensing than on methods development. A significant and worthwhile body of work exists using remote sensing to study water quality in complex near coast ocean environments as well as the Laurentian Great Lakes (see reviews [3–6]). While critical to the development of inland water quality remote sensing methods, this body of work was excluded from this review in order to better focus on lake, river, and estuary remote sensing applications and how those applications have changed over time. Similarly, studies using strictly in situ reflectance were excluded because our focus was on remote sensing from satellites or airborne platforms. The final subset was read to analyze overarching trends in research focus and scale. Each of the resulting 236 papers was subsequently classified into one of the four categories outlined below.

1. Purely methodological: The purpose of the paper is to present and validate a new model or methodology. Results consist of model validation and error metrics. No figures depicting spatial or temporal patterns are present.
2. Methodological with pattern analysis: The paper is predominately methods development and validation but includes some figures applying the proposed model either spatially or temporally.
3. Trend/pattern analysis: The purpose of the paper is to examine spatiotemporal patterns and/or trends in water quality within the study area, with trends defined has having directionality over space or time. Model validation results are presented for transparency, but the bulk of the results and discussion focusses on either spatial or temporal trend analysis. The preponderance of figures and tables depict maps, time-series, or other spatiotemporal analyses.
4. Water quality science research with a focus on impacts and drivers: The paper contains specific hypotheses and/or science questions to be directly addressed. Results and discussion focus on spatiotemporal dynamics of water quality as well as the drivers and/or impacts of changing water

quality. The preponderance of figures and tables present within the paper depict either trends or relationships between the parameter of interest and associated drivers/impacts.

Key questions that determined the classification of the papers included:

1. Is there a specific hypothesis or science question addressed?
2. Is there any spatial or temporal analysis of patterns or trends in the study area?
3. Are the majority of the figures and tables focused on validating a proposed model, or are they examining trends, drivers, and impacts of inland water quality?

With regards to the third criterion, figures and tables within each paper were categorized into the four groups depending on whether they provided background information, model validation, or spatiotemporal analysis (details in Table S3). The final index (Appendix A) depicts a field of research that has evolved, particularly in the last decade, from almost universally methods-focused into one in which new methodologies, data products, and increased computing power are creating opportunities to address science questions related to water quality in novel ways.

The overall trend in the publication counts of the detailed dataset closely parallels the power-law trend in the broad Scopus query, including a comparable spike in publications after 2008. Similarly, 75% of the studies resulting from the various searches focus on lakes and lake related water quality parameters (Figure 3). Eutrophication-associated parameters (chlorophyll, clarity, and cyanobacteria) are almost entirely measured in lake systems. In contrast, studies focusing on rivers, deltas, and estuaries are almost exclusively measuring sediment loading and transport parameters (TSS and turbidity).

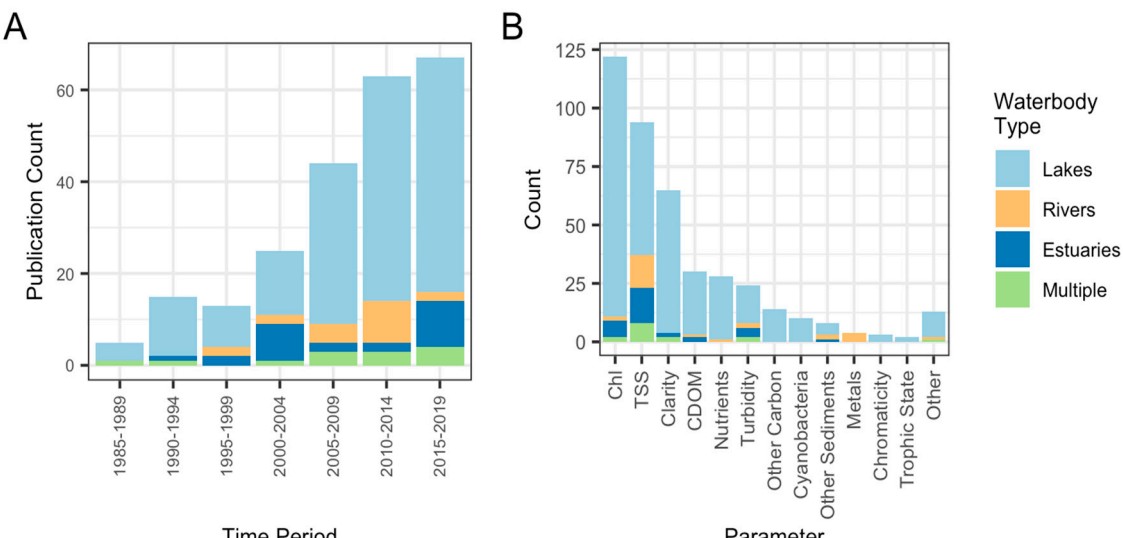

**Figure 3.** Publication counts within the detailed index binned by time (**A**) and water quality parameter of interest (**B**). Colors represent type of waterbody being researched.

In total, the included papers presented 411 models for constituent retrieval. Of these, only 70% reported some measure of goodness-of-fit or absolute error, and only 23% reported some measure of validation, with validation defined as an error metric derived from data not used in building the model. The most commonly reported metric was a coefficient of determination ($R^2$), with mean recorded values of 0.76 ($\sigma = 0.184$) for model fit and 0.79 ($\sigma = 0.159$) for model validation (Figure 4). Simple linear regression of $R^2$ values over time indicate that model fit has decreased ($p = 0.011$) and model validation has shown no significant trend ($p = 0.633$). However, more recent models frequently cover larger spatiotemporal domains and represent more difficult constituent retrieval, possibly leading to reduced model fit. While $R^2$ values are not the most robust stand-alone metric of model performance [215],

comparisons utilizing other common metrics are difficult due to the lack of standardization between reported metrics within the reviewed publications. In total, over 35 different error metrics were identified within the literature. Many of these represent differences in terminology as opposed to the actual statistical measure. For example, root mean square error (RMSE) is referred to in nine different ways in total, with variations both in terminology (e.g., root mean square error and root mean square deviation) and metric transformation (e.g., percent, normalized, relative, and log values). Similar ranges of variation occur for mean/median absolute error (MAE), standard error (SE), relative error (RE), and bias. This disparity in reporting measures makes it difficult to accurately compare model error across studies without significant burden on the reader. However, examination of the most common metric, $R^2$, suggests consistently strong model fits dating back to the 1970s (Figure 4). These results suggest that the potential has long existed for remote sensing to contribute to addressing scientific questions related to water quality. The reasons for the lag between methods development and scientific application remain uncertain. Two possible explanations are that the empirical models that dominate early literature were too site-specific to be useful at larger scales, or that perceptions of the usefulness of remote sensing in water quality research differed between the remote sensing community and fields like hydrology, limnology, and ecology.

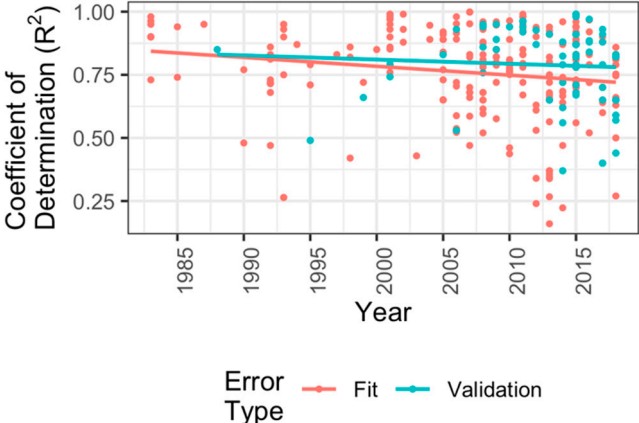

**Figure 4.** Reported $R^2$ values for model fit and model validation along with linear regressions of $R^2$ over time.

Trends in modelling approach indicate a fairly static field up until the early 1990s, with empirical modelling approaches comprising 50%–80% of all publications for nearly 20 years (Figure 5). The mid-2000s show an increase in publications employing machine learning models and pre-produced satellite products. The emergence and subsequent decline of product-based studies from 2008–2015 likely corresponds to the launch of the Medium Resolution Imaging Spectrometer (MERIS) in 2002 and its decommission in 2012. MERIS presented a unique step towards global products through the development of the BEAM processing toolbox (Brockman Consult in collaboration with the European Space Agency), which utilizes a neural network scheme to simultaneously conduct atmospheric correction and water quality estimates. BEAM provided ready-made water quality products to inland water and ocean researchers alike, though validation of the products was regionally inconsistent [216–218].

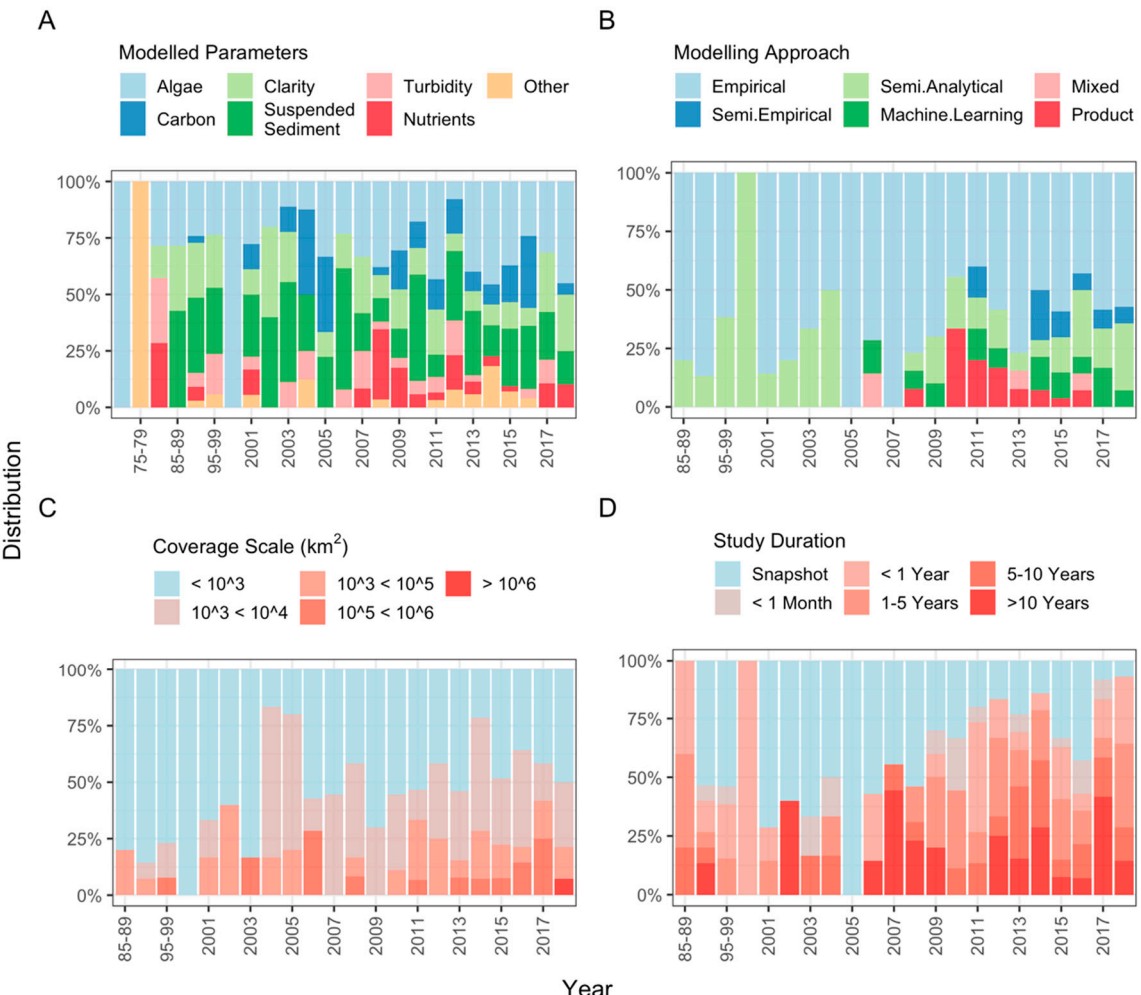

**Figure 5.** Temporal distributions of modelled parameters (**A**), modelling approach (**B**), study spatial scale (**C**), and study temporal scale (**D**). Results for 1975–1999 are reported in five-year windows due to the relatively small number of studies published during this time period.

The development of BEAM's neural network scheme and the rise in machine learning approaches starting around 2000 is likely attributable to increased computational capabilities and a proliferation of specialized software in common programming environments like R and Python. For the former, packages like Rpart [219], originally released in 1999, and nnet [220] created previously unavailable access to decision tree and neural network modelling approaches. For Python, software development throughout the 2000s led to comprehensive machine learning libraries such as Scikit-learn [221], which provided both access to common machine learning algorithms and a framework for their calibration and validation. These machine learning tools, among others, emerged in part due to an increased need for open source software that promoted study replicability, researcher access, and collaborative code development for machine learning researchers across fields [222].

The emergence of machine learning approaches in remote sensing of inland waters is paralleled by an increase in semi-empirical models. Initially, this late appearance of semi-empirical models appears unintuitive since they are computationally inexpensive and closely parallel older terrestrial indexes like NDVI; however, their emergence is likely explained by a proliferation of data from ocean color sensors such as SeaWiFs (launched 1997), MODIS Terra and Aqua (1999 and 2002 respectively), and MERIS (2002). With MERIS specifically, its high spectral resolution and chlorophyll-specific band centers allowed for better detection of absorption features and backscatter peaks that facilitate semi-empirical models [86]. However, due to their coarse spatial resolution, these studies are mostly limited to larger

lakes. These sensors were subsequently joined by the hyperspectral sensor Hyperion in 2000, which created new opportunities for semi-analytical water constituent retrieval [60].

The temporal trends described above show distinct spatial patterns with an overall dominance of studies located in the U.S., Europe, and China (Figure 6). China and the U.S., respectively, comprise 20% and 24% of the total studies included, with a notable clustering of long-term, large-scale studies in the Yangtze Basin. Spatiotemporal trends in publication dates depict a temporal expansion outward, with the earliest studies located almost exclusively in the U.S and subsequent publications spreading out across the globe. However, it should be noted that this trend may be partially attributable to a language bias in early publications where there is less access to non-English papers.

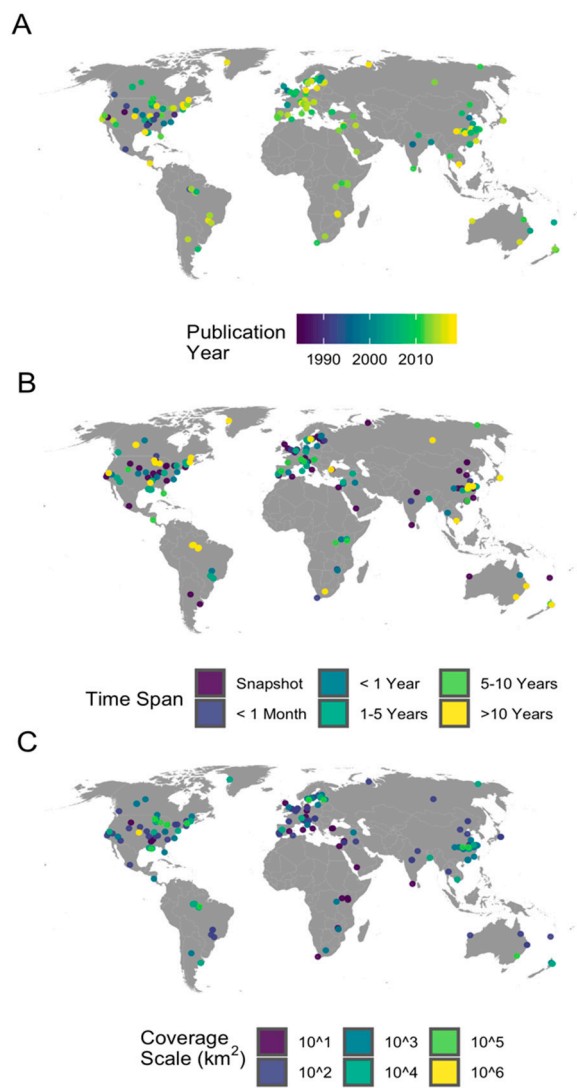

**Figure 6.** Spatial distribution of study publication date (**A**), timespan (**B**), and spatial scale (**C**).

## 6. From Methods to Applications: An Overview of Inland Water Remote Sensing

The study of water quality in lakes, rivers, and estuaries using remote sensing has expanded substantially over the past 50 years. When considering the *intent* of the publications as opposed to just the number, it is apparent that only in the past 10–15 years has inland water remote sensing consistently been used as a powerful analytical tool informing the broader inland water literature. In the papers reviewed for this analysis, twice as many studies were published in the past ten years as in the previous 28 years combined, a rate much faster than the growth of academic publishing as a whole.

Of papers published since 2008, nearly 30% focus on examining drivers and impacts of water quality, compared to only 7% for the period prior (Figure 7).

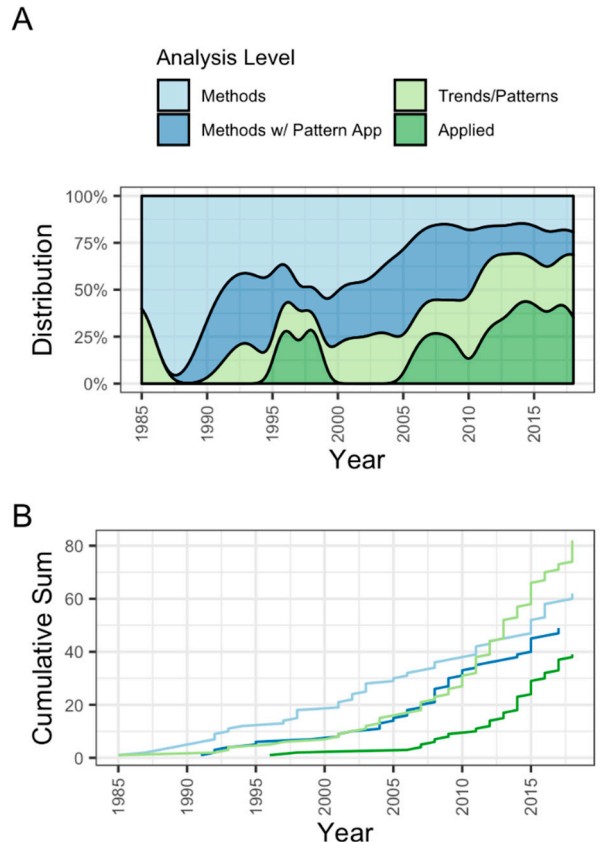

**Figure 7.** Distribution of publication focus through time (**A**). Cumulative sum totals (**B**) for methods categories (n = 113; 49%), trend/pattern papers (n = 81; 35%), and water quality science papers (n = 38; 16%) show decreasing dominance of methods over time.

Studies are also expanding into longer time-frames and over larger spatial scales (Figure 5). Pre-2008, the average study covered tens of square kilometers over a 2 year period. Post-2008, the average study examines hundreds of square kilometers over a period of 5 years. This expansion requires the caveat that study scale is used as a proxy for the number of distinct lakes, and some of the increase in study scale may result from an increase in the average lake size rather than the total number of optically unique waterbodies. Similarly, longer-term studies largely focus on simpler metrics such as water clarity and TSS, in part due to ongoing challenges modelling the more complex spectral signatures of chl-a and CDOM. Satellites like Landsat provide long time series of observations but lack the radiometric and spectral resolution to model more complex parameters. Pearson's Correlation Coefficients [223] were calculated to identify relationships between study scale, duration, publishing date, and category. The category variable was converted to a numeric (1–4) in order of level of analysis (1, methods development; 2, methods with pattern analysis; 3, trend/pattern analysis; 4, water quality science). While the categorical classification of the included papers is partially subjective, their correlations with other study parameters are still included to provide insight into how the scientific application of publications has changed with study scale and duration. The resulting correlation matrix (Table 2) depicts a clear pattern between study scale and impact over time. All four of the included variables were positively correlated at a 99% significance level (with the exception of study category and scale, $p = 0.014$). While none of the correlations are particularly strong, their significance and consistency indicate that studies published later tend to cover larger spatiotemporal domains and focus more on analyzing water quality dynamics and impacts than on methods development.

**Table 2.** Correlation matrix of key study parameters. All correlations are significant at a 99% confidence interval (***). Study category rescaled to 1–4, representing the four levels of analysis from purely methodological to water quality science papers.

|  | Pub. Year | Study Duration | Study Scale | Study Category |
|---|---|---|---|---|
| Pub. Year | 1 | 0.171 | 0.255 | 0.342 |
| Study Duration | *** | 1 | 0.326 | 0.32 |
| Study Scale | *** | *** | 1 | 0.173 |
| Study Category | *** | *** | *** | 1 |

This shift in study focus, scale, and duration all suggest that remote sensing is becoming a useful tool in understanding inland water quality rather than an area for methodological study among remote sensing specialists. Publications representative of this shift towards hypothesis-driven science vary significantly in their focus, with emphasis on hydrological processes, drivers of water quality, public hazard identification, and impacts of degraded water quality (Table S4).

Studies examining drivers of water quality at local to regional scales comprise the largest group of water quality science publications. Recent work has examined climatic, anthropogenic, and landscape-scale variables that interact with complex biogeophysical water quality properties. Work by Lymburner et al. (2016) [175] presents a 30 year analysis of TSS in Australian lakes, showing distinct relationships between El Niño Southern Oscillations and fluctuations in TSS levels. Olmanson, Brezonik, and Bauer (2014) [224], examined a 20 year record of remotely sensed water clarity for over 10,000 lakes in Minnesota. Their results showed significant differences in overall trends based on land use and eco-region. Ng et al. (2011) [225] and Curtarelli et al. (2015) [96] both incorporated remotely sensed chl-a data into hydrologic models and found that thermal stratification and mixing were key drivers of algal bloom growth and dispersion. Work done by Rose et al. (2017) [147] showed that controls on water clarity move from local to watershed scales during dry and wet years respectively. Other studies focusing on climatic drivers of water quality have used remote sensing to analyze the impacts of hurricanes [226], typhoons [227], and growing season length [228] on various water quality metrics.

Studies focusing on anthropogenic drivers have brought to light the impacts of human activities on freshwater resources for areas ranging from individual lakes to entire states. Work by Cui and others (2009, 2013) [229,230] examined the combined effects of precipitation, river flows, and dredging on TSS levels in Poyang Lake in China. They found that the combined precipitation and anthropogenic impacts degraded water quality far more than either individual driver could on its own. At a basin scale, Ren et al. (2018) [231] and Hou et al. (2017) [232] conducted studies examining how the Three Gorges Reservoir affected water clarity and TSS dynamics in the Yangtze Basin. In the Peace-Athabasca Delta, Pavelsky and Smith (2009) [233] and Long and Pavelsky (2013) [234] utilized multi-temporal images of sediment loads to calculate river velocity and recharge for floodplain lakes.

Studies using remote sensing of water quality to address scientific questions extend beyond the field of hydrology and into biology and public health. Sandström et al. (2016) [235] utilized remotely sensed CDOM and chl-a concentrations to analyze fish habitat assemblages and biodiversity. For public health, inland water remote sensing is helping to analyze disease distribution and drinking water hazards. Fichot et al. (2016) [68] identified spatial patterns of methylmercury in the San Francisco Bay area using an airplane mounted hyperspectral sensor. Qin et al. (2015) [27] developed a dynamic forecasting model capable of predicting the presence of toxic algal blooms. The model ultimately resulted in over one million tons of algal scum being removed from a drinking water reservoir in China [27]. Other authors have similarly identified public threats to drinking water in Lake Mead (USA) [182] and Lake Chaohu in China [190]. Two specific studies stood out through their novel use of remote sensing to facilitate epidemiological studies. Torbick et al. (2014) [236] incorporated Landsat-derived water quality parameters into an eco-epidemiological model to examine the distribution of amyotrophic lateral sclerosis (ALS) across New England. They found that close

proximity to waterbodies with elevated levels of nitrogen increased the odds of being located within an ALS hotspot by 167%. Similarly, Finger et al. (2014) [237] incorporated remotely sensed chl-a measurements into a model of cholera dynamics within the Democratic Republic of Congo.

One additional subset of the reviewed literature merits discussion when considering advances in the field; specifically, researchers who are continuing to expand the spatiotemporal scale of their study areas. The need for global data products has received increasing attention in recent years as an essential aspect to protecting threatened freshwater resources [15–17]. Within the U.S., work towards this goal includes state-wide analyzes of Secchi Disk depth in Minnesota [156,224] and Maine [238], and a national approach to modelling lake chl-a [188]. Outside the U.S., previously mentioned work by Lymberner et al. (2016) [175] in Australian lakes and Hou et al. (2017) [232] in the Yangtze basin both cover areas of tens of thousands of square kilometers, albeit without including every lake in the study region. At a global level, Ho et al. (2019) [239] recently analyzed the prevalence of harmful algal blooms in 71 lakes over the span of three decades, further increasing the spatiotemporal domain of inland water remote sensing.

Publications like those mentioned above are complemented by a host of living databases and interactive web services that are increasing access to near real-time water quality information. The Copernicus Inland Water Service provides semi-continuous (2002–2012, 2016-present) turbidity and chl-a observations for approximately 1000 of the world's largest lakes (https://land.copernicus.eu/global/products/lwq). Similarly, the Minnesota LakeBrowser provides periodic measurements of chl-a, CDOM, and water clarity dating back to 2002 for over 10,000 lakes across the state (https://lakes.rs.umn.edu/). These publicly available databases are being supplemented by private companies like EOMAP (https://www.eomap.com/) which provide remotely sensed estimates of water quality parameters on a contract basis around the globe. While validation of some of these products is difficult to obtain, they are facilitating increased access to water quality data for water managers and researchers alike. Improvements in modeling methodologies and growing access to both in situ and earth observation data are setting the stage for future studies at larger and larger scales.

## 7. Emerging Trends in the Remote Sensing of Water Quality

The past decade has seen a dramatic growth in the resources necessary to remotely sense inland water quality. One example highlighted here is the 2008 shift to open access Landsat data—after which, publication counts rose and study scale and duration increased significantly. However, the Landsat archive is only one of numerous petabyte-size archives of earth observation data provided by government agencies such as NASA, the USGS, NOAA, and the European Space Agency. These archives are constantly expanding and will continue to do so in the coming years. Starting in 2010, access to these data sources further increased with the release of the Google Earth Engine platform, which hosts imagery and resulting data products from over a dozen different earth observation sensors. The platform provides free access to these datasets along with cloud-based processing, dramatically increasing the computational power of remote sensing researchers across fields. For inland water remote sensing, Lin et al. (2018) [188] combined in situ data from the 2007 National Lake Assessment (N = 1157 lakes) with Landsat data and machine learning algorithms built into Google Earth Engine to develop a well-validated national model for lake chl-a (RMSE = 34.9 μg/L). Similarly, Overeem et al. (2017) [110] used Google Earth Engine to model sediment export from Greenland over 14 years. Today, the platform continues to grow and increase in usefulness, adding approximately 6000 scenes daily from various active satellite missions, with a latency of approximately 24 h [240]. The power of Google Earth Engine essentially provides researchers with supercomputing capabilities from their local machines, dramatically increasing the scales at which earth observation research can take place. Platforms like Google Earth Engine are complimented by an ever-growing body of processing and analysis software in common programming languages like R [241].

While the provision of open-access satellite imagery to researchers is essential to the progression of the field, it alone cannot account for the shift in research focus and scale outlined above. Paralleling the

rise in remote sensing data availability over the past decade has been a rise in the in situ data available for model calibration and validation. In the past, the burden of collecting this data frequently fell on individual researchers, significantly limiting the amount of field data available. Recent databases provided by government agencies, NGOs, and researchers alike are providing a wealth of freely available in situ data that are easily accessible. At a global level, the GEMStat database maintained through the International Centre for Water Resources and Global Change, provides over 4 million observations of lakes, rivers, wetlands, and groundwater systems from 4000 sites spread over 75 countries (https://gemstat.org/). In the U.S., The National Water Quality Portal (WQP), released in 2012 by the USGS, EPA, and National Water Quality Monitoring Network, provides national coverage of archived state, federal, and tribal water quality field measurements. In total it assimilates and standardizes monitoring data for over 2 million individual sampling sites [242]. The Lake Multi-Scaled Geospatial and Temporal Database (LAGOS-NE) provides a similar assimilation of in situ water quality measurements for 17 water-rich states in the upper Midwest and Northeast United States, providing historical field data for over 51,000 lakes and reservoirs [243]. These datasets have already been used as calibration and validation data for remote sensing of water skin temperature [244]. In Europe, national-scale water quality data for inland and coastal waters are compiled from participating agencies into the Waterbase dataset, which is harmonized and made research ready under the WISE system (water information system for Europe) [245]. These official data sources can be supplemented with novel collections aggregated through citizen science campaigns. These include Eye On Water (http://www.eyeonwater.org/) and Seen-monitoring (http://www.seen-transparent.de/) in Europe, the Secchi-Dip In in North America (http://www.secchidipin.org/), and state level efforts in Minnesota, Wisconsin, Michigan, and Maine (https://www.pca.state.mn.us/water/citizen-water-monitoring, https://www.uwsp.edu/cnr-ap/UWEXLakes/Pages/programs/clmn/default.aspx, https://micorps.net/lake-monitoring/, and https://www.lakestewardsofmaine.org/ respectively). Together these campaigns have collected hundreds of thousands of observations available to researchers. The new AquaSat database from Ross et al. (2019) [246] uses Google Earth Engine to extract coincident (+/−1 day) Landsat reflectance values for in situ measurements found in the WQP and LAGOS-NE. The result is the first dataset of its kind, providing over 500,000 paired observations of reflectance values and associated water quality parameters in optically complex waters dating back to 1984. Databases such as these provide data continuity, cost and time savings for researchers, and large calibration and validation samples for model development.

The development and expansion of new and existing databases is paralleled by the development of new sensor technology. Airborne hyperspectral sensors capable of capturing contiguous spectral signatures of water-leaving radiance have provided new levels of precision to measure optically active constituents (see reviews by [3,58]). These airborne campaigns are working towards satellite missions such as NASA's Surface Biology and Geology mission (SBG, in development), Italy's PRecursore IperSpettrale della Missione Applicativa (PRISMA, launched 22 March 2019), Japan's Hyperspectral Imaging Suite (HISUI, planned 2019), and Germany's Environmental Mapping and Analysis Program (EnMAP, planned 2020) [247]. These spaceborne imaging spectrometers will increase spatiotemporal transferability of retrieval models, improve overall constituent retrieval, facilitate biogeochemical composition analysis, enable benthic habitat identification in optically shallow water bodies, and allow for the retrieval of additional detectable water quality parameters that are currently unfeasible with broadband, multispectral sensors, all while providing global hyperspectral data at roughly 30 m resolution [17,18]. Traditional governmental satellite missions are being supplemented with a host of novel earth observation technologies being developed by commercial companies such as Planet (https://www.planet.com/), MAXAR (https://www.maxar.com/), and Airbus (https://www.airbus.com/). These private platforms are creating novel opportunities for hydrological remote sensing through public and academic research partnerships. For example, Planet, which operates over 150 small imaging satellites that provide daily global imagery at 3–5 m resolution, collaborated with Cooley et al. (2017) [248] to study lake connectivity in the Yukon Flats region of Alaska at previously unfeasible

spatial scales. For inland water quality, the high spatial and temporal resolution of such satellite constellations will allow for detection of short-term phenomena like algal blooms in streams and lakes that are currently too small to study with publicly available satellite imagery. These efforts to improve research in small aquatic systems are being further aided by the increased use of unmanned aerial vehicles and even smartphones [249].

These emerging technologies will allow the inland water quality remote sensing community to overcome historic challenges and examine new science questions. However, this process will require dedicated researchers and reliable funding sources. While emerging technologies hold promise, they also present new challenges. Hyperion, the first spaceborne hyperspectral sensor believed to be appropriate for inland waters, showed initial promise [60] but ultimately proved unreliable over waterbodies due to its low signal to noise ratio and radiometric instability [250]. The Planet constellation of CubeSats, while providing unprecedented spatial and temporal resolution, are subject to geolocation accuracy errors and inconsistencies in radiometric calibration between satellites [248]. These issues are in addition to well-characterized challenges including robust atmospheric correction and solving adjacency effects, both of which need to be applied across sensors to create comparable datasets. Solutions to these existing challenges will likely be developed through improvements in sensor engineering, computational capacity, and modelling approaches, as well as growing collaborative efforts by international groups such as IOCCG [20,251] and the Committee on Earth Observation Satellites [54]. As existing issues are overcome, remote sensing of inland water quality can be applied to address relevant scientific questions and conservation goals, including those outlined in the National Research Council Decadal Survey [252] and the EU Water Framework Directive [253]. Conducting such research will help solve water quality issues of global importance and better inform water managers, policy makers, and the scientific community regarding critical science questions. Some of the most pressing questions synthesized from the reviewed literature include:

- How does biogeochemical cycling of suspended sediments and CDOM in lakes and rivers contribute to the global carbon cycle?
- How are added nutrient inputs and warming air temperatures contributing to the frequency and distribution of harmful algal blooms in lakes and reservoirs?
- What is the impact of anthropogenic development, including urbanization and reservoir construction, on basin-wide water quality?
- What are the patterns and trends in the biogeochemistry of water resources in remote, vulnerable areas including the arctic and boreal regions?
- How are changes in water quality affecting the biological structure of freshwater resources at regional to global scales?
- How are changing water quality dynamics impacting important drinking water resources?

## 8. Conclusions

The bibliometric analysis presented here highlights the dramatic growth of inland water quality remote sensing studies, far outpacing the average rate of increase in academic publishing as a whole. The past 50 years have produced hundreds of remote sensing publications accurately estimating biogeochemical water quality parameters; however, the majority of these focus on methods development rather than using remote sensing as a tool to better understand inland water quality dynamics. Detailed examination of 236 of the most relevant publications returned by search queries indicates that the past 10–15 years has brought about a focal shift within the field, where researchers are moving beyond methods development towards research focused on spatiotemporally explicit water quality dynamics. This shift is partially attributable to the development of new satellite and in situ datasets, improved access to satellite imagery, and increased computational/software capabilities. The current change in focus within the field is similar in nature to the shift that occurred in ocean color and terrestrial remote sensing throughout the 1980s and 1990s—after which, both fields applied remote sensing to answer

some of the most pressing science questions of their time. For inland water quality, the progression of research is evidenced by a subset of recent publications which have begun to leverage remote sensing to examine water quality trends, ecological and anthropogenic drivers, and resulting impacts of changing water quality on ecosystem function and water resources. This shift has been accompanied by a significant increase in the spatiotemporal scale of analysis, moving the field closer to providing national to global-scale data products for policy makers, water managers, and scientists. The increase in high quality science and study scale within the field continues to be facilitated by improved datasets and growing computational capacity. New data products like AquaSat [246] promise to continue this trajectory of growth and facilitate a new generation of inland water remote sensing research.

Based on the literature reviewed here, future inland water quality remote sensing work will benefit greatly from the following recommendations:

- Continued development of generalizable constituent retrieval models, including atmospheric corrections, that are applicable across large spatiotemporal domains and across differing sensors.
- The expanded application of robust, generalizable models to better understand global processes including erosion and deposition, terrestrial carbon and nutrient cycling, and trends in algal bloom dynamics in inland waters.
- Improved communication between experts in remote sensing and scientists in fields such as hydrology, limnology, and ecology in order to facilitate the wider adoption of remote sensing models in scientific studies of water quality.
- The development of user-friendly tools that inform local water managers of remotely sensed changes in water quality to promote sound policy and the conservation of essential freshwater resources.

**Supplementary Materials:** The following are available online at http://www.mdpi.com/2073-4441/12/1/169/s1. Table S1: Summary of the optically active water quality indicators measured through inland water remote sensing studies. Specific studies, sensor information, and modelling approaches focusing on the listed parameters and included in the review can be accessed through the inland water quality remote sensing index linked to in Appendix A. For a more detailed, technical discussion of specific algorithms and spectral responses for each parameter, see Matthews (2011) [4], Gholizadeh et al. (2016) [3], and Giardino et al. (2019) [18]. Table S2: Summary of the common approaches to algorithm development for remote sensing of inland waters studies. Specific studies using the described approaches and included in the review can be accessed through the inland water quality remote sensing index linked to in Appendix A. Table S3: Summary of collected information for the detailed literature review index. Table S4: Summary of studies using remote sensing to analyze impacts and drivers of water quality and classified as water quality science papers within the analysis.

**Author Contributions:** Conceptualization and methodology conducted by S.N.T., M.R.V.R., and T.M.P. Formal analysis done by S.N.T., M.R.V.R., and D.J. Writing and draft preparation done by S.N.T. Review and editing done by S.N.T., M.R.V.R., T.P., D.J., and M.S. All authors contributed substantially to the produced work. All authors have read and agreed to the published version of the manuscript.

**Funding:** Support for the manuscript was provided by NASA NESSF 80NSSC18K1398. Part of this work was conducted at the Jet Propulsion Laboratory, California Institute of Technology, under a contract with the National Aeronautics and Space Administration.

**Acknowledgments:** We are grateful to all those who provided feedback on this manuscript.

**Conflicts of Interest:** The authors declare no conflict of interest.

## Appendix A

The inland water quality remote sensing index can be found in its entirety here: https://docs.google.com/spreadsheets/d/1GMka4B-E16FmXBWjv0lhBN0T-07oeZT4riepMMHvZz4/edit?usp=sharingusp=sharing; the code and data for all analysis and figures can be found here: https://github.com/SimonTopp/rs.iw.review.

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
