# Peer review of "Research Trends in the Use of Remote Sensing for Inland Water Quality Science: Moving Towards Multidisciplinary Applications"

_water, doi:10.3390/w12010169_

Round 1

Reviewer 1 Report

Dear editor:

   Sorry for the delay in reviewing this manuscript. Now I have completed the reviewing. Overall, it is a nice review manuscript involved with the inland water quality monitoring by remote sensing. However, before the acceptance of this manuscript, the reviewer still has several questions for the authors:

Line 71: Why the Laurentian Great Lakes were excluded from the review? I think there must be many studies involved with water quality monitoring through remote sensing in this region. Line 87-127: Could you please make a table or a figure to list the major sensors used for the water quality monitoring, including the comparison of the spatial, spectral, temporal, radiometric resolutions? The context does not provide detailed information on the sensors. Section 2.1-2.4: Could you please provide a table to summarize the four indicators (parameters) used for the monitoring, including the definition, the importance, the number of paper involved, the specific bands or method used? Section 3.1-3.3: Please include a table summarizing the detail of the methods used in remote sensing, including the number of papers involved in the studies. Section 4: This part is unclear, the limitations and challenges should come from all the previous studies you reviewed, but this section lacks logic for the connections between previous studies and the limitations. Line 442 and 830: Why do you have two Table 1 in one manuscript? The table style is not quite right for the scientific research paper, please check all the styles. Line 823-824: Please check the link of the Google spreadsheets, I tried to open it but the file does not exist anymore. Line: 516: I like Figure 4, but did you count the paper used similar parameters (e.g. the ‘r’ instead of ‘R2’)? Line 463-441: I think it is good to know that the free release of the Landsat archive contributes a lot for the water quality remote sensing. But the specific design of TM, ETM+, and OLI are not focusing on the water body, especially for the bandwidth and the SNR. Line 446: Figure 2, could you make the size uniform for both subplots in this figure? Also, you need to label individual subplots in one figure. Line 620 and 832: Again, you had two Table 2 in on manuscript. The conclusion part needs bullet to highlight the findings.

Reviewer 2 Report

The paper “Research trends in the use of remote sensing for inland water quality science: moving towards multidisciplinary applications” is interesting and focus on the more relevant issues of area. I only have specific comments.

Specific comments

Page 3, line 129. Here solids is better than sediments because organic suspended sediments sounds inappropriate.

Page 4, line 154. Peaks do not seem appropriated because peaks refers to something high and absorption in reflectance spectra is low. Maybe feature will be better. Please check this in the whole text.

Page 4, line 164. TSS appears here for the first time. It still need to be cited along with its meaning, likely, in the beginning of this paragraph.

Page 10, Figure 1. Call Figure 1 in the text before figure. Please check this in the whole text. Figure 3 is ok.

Page 11, Table 1. Include Figures and Tables after call them in the text. Table 2 is ok.

Page 12, Figure 2. Indicate different graphs using letters: (a) and (b)

Page 17, lines 574-580. After I understood this paragraph, but at first this paragraph do not look linked to above paragraphs of this subsection. It will be appropriated to use a connection.

Page 19, line 603. “radiometric resolution of satellites like Landsat that provide longer time series of observations” is confuse: ‘radiometric resolution’ and ‘temporal’.

Page 21, lines 692 and 698. It is Earth Engine or ‘Google Earth Engine’.

Page 23, line 768. What it means NRC?
